# LEARNING TO MIX $n$-STEP RETURNS: GENERALIZING $\lambda$-RETURNS FOR DEEP REINFORCEMENT LEARNING

## ABSTRACT

Reinforcement Learning (RL) can model complex behavior policies for goal-directed sequential decision making tasks. A hallmark of RL algorithms is Temporal Difference (TD) learning: value function for the current state is moved towards a bootstrapped target that is estimated using the next state's value function. $\lambda$-returns define the target of the RL agent as a weighted combination of rewards estimated by using multiple many-step look-aheads. Although mathematically tractable, the use of exponentially decaying weighting of $n$-step returns based targets in $\lambda$-returns is a rather ad-hoc design choice. Our major contribution is that we propose a generalization of $\lambda$-returns called *Confidence-based Autodidactic Returns* (CAR), wherein the RL agent learns the weighting of the $n$-step returns in an end-to-end manner. In contrast to $\lambda$-returns wherein the RL agent is restricted to use an exponentially decaying weighting scheme, CAR allows the agent to learn to decide how much it wants to weigh the $n$-step returns based targets. Our experiments, in addition to showing the efficacy of CAR, also empirically demonstrate that using sophisticated weighted mixtures of multi-step returns (like CAR and $\lambda$-returns) considerably outperforms the use of $n$-step returns. We perform our experiments on the Asynchronous Advantage Actor Critic (A3C) algorithm in the Atari 2600 domain.

## 1 INTRODUCTION

Reinforcement Learning (RL) (Sutton & Barto, 1998) is often used to solve goal-directed sequential decision making tasks wherein conventional Machine Learning methods such as supervised learning are not suitable. Goal-directed sequential decision making tasks are modeled as Markov Decision Process (MDP) (Puterman, 2014). Traditionally, tabular methods were extensively used for solving MDPs wherein value function or policy estimates were maintained for every state. Such methods become infeasible when the underlying state space of the problem is exponentially large or continuous. Traditional RL methods have also used linear function approximators in conjunction with hand-crafted state spaces for learning policies and value functions. This need for hand-crafted task-specific features has limited the applicability of RL, traditionally.

Recent advances in representation learning in the form of deep neural networks provide us with an effective way to achieve generalization (Bengio et al., 2009; LeCun et al., 2015). Deep neural networks can learn hierarchically compositional representations that enable RL algorithms to generalize over large state spaces. The use of deep neural networks in conjunction with RL objectives has shown remarkable results such as learning to solve the Atari 2600 tasks from raw pixels (Bellemare et al., 2013; Mnih et al., 2015; 2016; Sharma et al., 2017; Jaderberg et al., 2017), learning to solve complex simulated physics tasks (Todorov et al., 2012; Schulman et al., 2015a; Lillicrap et al., 2015) and showing super-human performance on the ancient board game of Go (Silver et al., 2016; 2017). Building accurate and powerful (in terms of generalization capabilities) state and action *value function* (Sutton & Barto, 1998) estimators is important for successful RL solutions. This is because many practical RL solutions (Q-Learning (Watkins & Dayan, 1992), SARSA (Rummery & Niranjan, 1994) and Actor-Critic Methods (Konda & Tsitsiklis, 2000)) use Temporal Difference (TD) Learning (Sutton, 1988). In TD learning, a $n$-step return is used as an estimate of the value function by means of bootstrapping from the $n^{th}$ state's value function estimate. On the other hand, in Monte Carlo learning, the cumulative reward obtained in the entire trajectory

following a particular state is used as an estimate for the value function of that state. The ability to build better estimates of the value functions directly results in better policy estimates as well as faster learning. $\lambda$-returns (LR) (Sutton & Barto, 1998) are very effective in this regard. They are effective for faster propagation of delayed rewards and also result in more reliable learning. LR provide a trade-off between using complete trajectories (Monte Carlo) and bootstrapping from n-step returns (TD learning). They model the TD target using a mixture of $n$-step returns, wherein the weights of successively longer returns are exponentially decayed. With the advent of deep RL, the use of multi-step returns has gained a lot of popularity (Mnih et al., 2016). However, it is to be noted that the use of exponentially decaying weighting for various $n$-step returns seems to be an ad-hoc design choice made by LR. In this paper, we start off by extensively benchmarking $\lambda$-returns (our experiments only use truncated $\lambda$-returns due to the nature of the DRL algorithm (A3C) that we work with and we then propose a generalization called the *Confidence-based Autodidactic Returns (CAR)*, In CAR, the DRL agent learns in an end-to-end manner, the weights to assign to the various $n$-step return based targets. Also in CAR, it's important to note that the weights assigned to various $n$-step returns change based on the different states from which bootstrapping is done. In this sense, CAR weights are dynamic and using them represents a significant level of sophistication as compared to the usage of $\lambda$-returns.

In summary, our contributions are:

1. To alleviate the need for some ad-hoc choice of weights as in the case of $\lambda$-returns, we propose a generalization called *Autodidactic Returns* and further present a novel derivative of it called *Confidence-based Autodidactic Returns (CAR)* in the DRL setting.

2. We empirically demonstrate that using sophisticated mixtures of multi-step return methods like $\lambda$-returns and Confidence-based Autodidactic Returns leads to considerable improvement in the performance of a DRL agent.

3. We analyze how the weights learned by CAR are different from that of $\lambda$-returns, what the weights signify and how they result in better estimates for the value function.

## 2 BACKGROUND

In this section, we present some basic concepts required to understand our work.

### 2.1 PRELIMINARIES

An MDP (Puterman, 2014) is defined as the tuple $\langle \mathcal{S}, \mathcal{A}, r, \mathcal{P}, \gamma \rangle$, where $\mathcal{S}$ is the set of states in the MDP, $\mathcal{A}$ is the set of actions, $r : \mathcal{S} \times \mathcal{A} \mapsto \mathbb{R}$ is the reward function, $\mathcal{P} : \mathcal{S} \times \mathcal{A} \times \mathcal{S} \mapsto [0, 1]$ is the transition probability function such that $\sum_{s'} p(s, a, s') = 1, \quad p(s, a, s') \geq 0$, and $\gamma \in [0, 1)$ is the discount factor. We consider a standard RL setting wherein the sequential decision-making task is modeled as an MDP and the agent interacts with an environment $\mathcal{E}$ over a number of discrete time steps. At a time step $t$, the agent receives a state $s_t$ and selects an action $a_t$ from the set of available actions $\mathcal{A}$. Given a state, the agent could decide to pick its action stochastically. Its policy $\pi$ is in general a mapping defined by: $\pi : \mathcal{S} \times \mathcal{A} \mapsto [0, 1]$ such that $\sum_{a \in \mathcal{A}} \pi(s, a) = 1, \quad \pi(s, a) \geq 0 \ \forall s \in \mathcal{S}, \ \forall a \in \mathcal{A}$. At any point in the MDP, the goal of the agent is to maximize the return, defined as: $G_t = \sum_{k=0}^{\infty} \gamma^k r_{t+k}$ which is the cumulative discounted future reward. The state value function of a policy $\pi$, $V^\pi(s)$ is defined as the expected return obtained by starting in state $s$ and picking actions according to $\pi$.

### 2.2 ACTOR CRITIC ALGORITHMS

Actor Critic algorithms (Konda & Tsitsiklis, 2000) are a class of approaches that directly parameterize the policy (using an *actor*) $\pi_{\theta_a}(a|s)$ and the value function (using a critic) $V_{\theta_c}(s)$. They update the policy parameters using Policy Gradient Theorem (Sutton et al., 1999; Silver et al., 2014) based objective functions. The value function estimates are used as baseline to reduce the variance in policy gradient estimates.

### 2.3 ASYNCHRONOUS ADVANTAGE ACTOR CRITIC

Asynchronous Advantage Actor Critic(A3C) (Mnih et al. (2016)) introduced the first class of actor-critic algorithms which worked on high-dimensional complex visual input space. The key insight in this work is that by executing multiple actor learners on different threads in a CPU, the RL agent can explore different parts of the state space simultaneously. This ensures that the updates made to the parameters of the agent are uncorrelated.

The actor can improve its policy by following an unbiased low-variance sample estimate of the gradient of its objective function with respect to its parameters, given by:

$$\nabla_{\theta_a} \log \pi_{\theta_a}(a_t|s_t)(G_t - V(s_t))$$

In practice, $G_t$ is often replaced with a biased lower variance estimate based on multi-step returns. In the A3C algorithm $n$-step returns are used as an estimate for the target $G_t$, where $n \leq m$ and $m$ is a hyper-parameter (which controls the level of rolling out of the policies). A3C estimates $G_t$ as:

$$G_t \approx \hat{V}(s_t) = \sum_{i=1}^{n} \gamma^{i-1} r_{t+i} + \gamma^n V(s_{t+n})$$

and hence the objective function for the actor becomes:

$$L(\theta_a) = \log \pi_{\theta_a}(a_t|s_t)\delta_t$$

where $\delta_t = \sum_{i=t}^{t+j-1} \gamma^{i-t} r_i + \gamma^j V(s_{t+j}) - V(s_t)$ is the $j$-step returns based TD error.

The critic in A3C models the value function $V(s)$ and improves its parameters based on sample estimates of the gradient of its loss function, given as: $\nabla_{\theta_c}(\hat{V}(s_t) - V_{\theta_c}(s_t))^2$.

### 2.4 WEIGHTED RETURNS

Weighted average of $n$-step return estimates for different $n$'s can be used for arriving at TD-targets as long as the sum of weights assigned to the various $n$-step returns is 1 (Sutton & Barto (1998)). In other words, given a weight vector $w = \left(w^{(1)}, w^{(2)}, \cdots, w^{(h)}\right)$, such that $\sum_{i=1}^{h} w^{(i)} = 1$, and $n$-step returns for $n \in \{1, 2, \cdots, h\}$: $G_t^{(1)}, G_t^{(2)}, \cdots, G_t^{(h)}$, we define a weighted return as

$$G_t^w = \sum_{n=1}^{h} w^{(n)} G_t^{(n)} \tag{1}$$

Note that the $n$-step return $G_t^{(n)}$ is defined as:

$$G_t^{(n)} = \sum_{i=1}^{n} \gamma^{i-1} r_{t+i} + \gamma^n V(s_{t+n}) \tag{2}$$

### 2.5 $\lambda$-RETURNS

A special case of $G_t^w$ is $G_t^\lambda$ (known as $\lambda$-returns) which is defined as:

$$G_t^\lambda = (1 - \lambda) \sum_{n=1}^{h-1} \lambda^{n-1} G_t^{(n)} + \lambda^{h-1} G_t^{(h)} \tag{3}$$

What we have defined here are a form of truncated $\lambda$-returns for TD-learning. These are the only kind that we experiment with, in our paper. We use truncated $\lambda$-returns because the A3C algorithm is designed in a way which makes it suitable for extension under truncated $\lambda$-returns. We leave the problem of generalizing our work to the full $\lambda$-returns as well as eligibility-traces ($\lambda$-returns are the forward view of eligibility traces) to future work.

## 3 $\lambda$-Returns and Beyond: Autodidactic Returns

### 3.1 Autodidactic Returns

*Autodidactic returns* are a form of weighted returns wherein the weight vector is also *learned* alongside the value function which is being approximated. It is this generalization which makes the returns *autodidactic*. Since the autodidactic returns we propose are constructed using weight vectors that are state dependent (the weights change with the state the agent encounters in the MDP), we denote the weight vector as $w(s_t)$. The autodidactic returns can be used for learning better approximations for the value functions using the TD(0) learning rule based update equation:

$$V_t(s_t) \leftarrow V_t(s_t) + \alpha \left( G_t^{w(s_t)} - V_t(s_t) \right) \tag{4}$$

In contrast with autodidactic returns, $\lambda$-returns assign weights to the various $n$-steps returns which are constants given a particular $\lambda$. We reiterate that the weights assigned by $\lambda$-returns don't change during the learning process. Therefore, the *autodidactic returns* are a generalization and assign weights to returns which are dynamic by construction. The autodidactic weights are learned by the agent, using the reward signal it receives while interacting with the environment.

### 3.2 Confidence-based Autodidactic Returns

All the $n$-step returns for state $s_t$ are estimates for $V(s_t)$ bootstrapped using the value function of corresponding $n^{th}$ future state $(V(s_{t+n}))$. But all those value functions are estimates themselves. Hence, one natural way for the RL agent to weigh an $n$-step return $\left( G_t^{(n)} \right)$ would be to compute this weight using some notion of *confidence* that the agent has in the *value function* estimate, $V(s_{t+n})$, using which the $n$-step return was estimated. The agent can weigh the $n$-returns based on how *confident* it is about bootstrapping from $V(s_{t+n})$ in order to obtain a good estimate for $V(s_t)$. We denote this *confidence* on $V(s_{t+n})$ as $c(s_{t+n})$. Given these *confidences*, the weight vector $w(s_t)$ can computed as:

$$w(s_t) = \left( w(s_t)^{(1)}, w(s_t)^{(2)}, \cdots, w(s_t)^{(m)} \right)$$

where $w(s_t)^{(i)}$ is given by:

$$w(s_t)^{(i)} = \frac{e^{c(s_{t+i})}}{\sum_{j=1}^{j=m} e^{c(s_{t+j})}} \tag{5}$$

The idea of weighing the returns based on a notion of confidence has been explored earlier (White & White, 2016; Thomas et al., 2015). In these works, learning or adapting the lambda parameter based on a notion of confidence/certainty under the bias-variance trade-off has been attempted, but the reason why only a few successful methods have emerged from that body of work is due to the difficult of quantifying, measuring and optimizing this certainty metric. In this work, we propose a simple and robust way to model this and we also address the question of what it means for a particular state to have a high value of *confidence* and how this leads to better estimates of the value function.

### 3.3 Using $\lambda$-returns in A3C

$\lambda$-returns have been well studied in literature (Peng & Williams, 1996; Sutton & Barto, 1998; Seijen & Sutton, 2014) and have been used in DRL setting as well (Schulman et al., 2015b; Gruslys et al., 2017). We propose a straightforward way to incorporate (truncated) $\lambda$-returns into the A3C framework. We call this combination as LRA3C.

The critic in A3C uses n-step return for arriving at good estimates for the value function. However, note that the TD-target can in general be based on any $n$-step return (or a mixture thereof). The A3C algorithm in specific is well suited for using weighted returns such as $\lambda$-returns since the algorithm already uses $n$-step return for bootstrapping. Using eqs. (1) to (3) makes it very easy to incorporate weighted returns into the A3C framework. The sample estimates for the gradients of the actor and the critic respectively become:

$$\nabla_{\theta_a} \log \pi_{\theta_a}(a_t|s_t)(G_t^\lambda - V(s_t))$$

$$\nabla_{\theta_c}(G_t^\lambda - V_{\theta_c}(s_t))^2$$

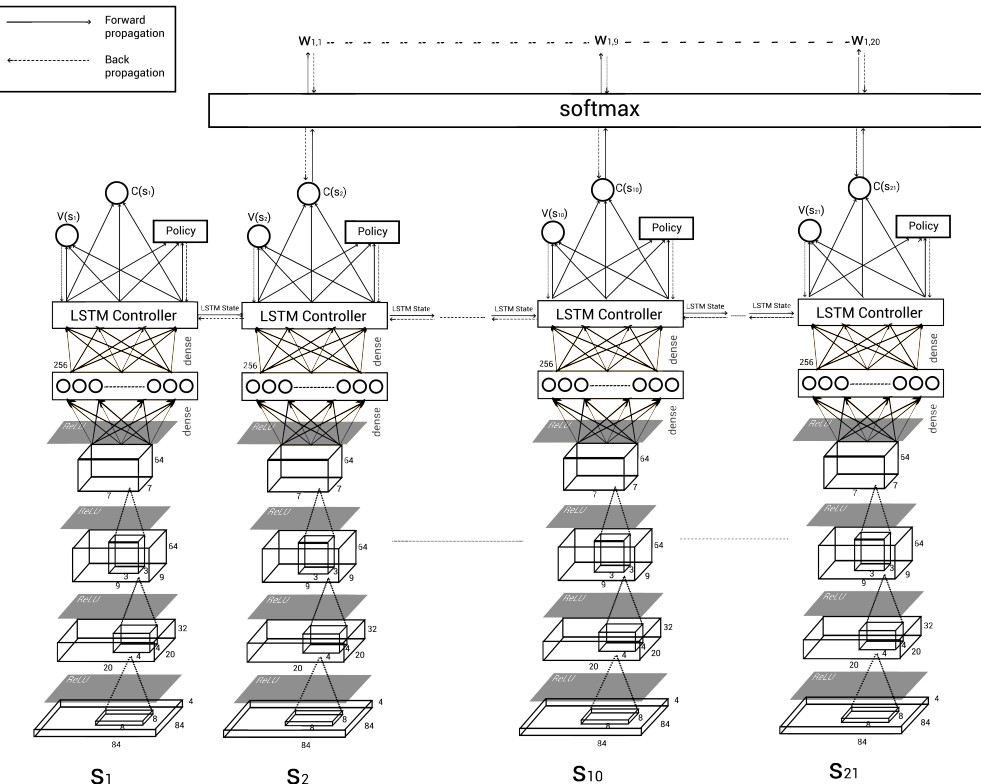

Figure 1: CARA3C network - Confidence-based weight vector calculation for state $s_1$.

### 3.4 USING AUTODIDACTIC RETURNS IN A3C

We propose to use *autodidactic returns* in place of normal $n$-step returns in the A3C framework. We call this combination as CARA3C. In a generic DRL setup, a forward pass is done through the network to obtain the value function of the current state. The parameters of the network are progressively updated based on the gradient of the loss function and the value function estimation (in general) becomes better. For predicting the confidence values, a distinct neural network is created which shares all but the last layer with the value function estimation network. So, every forward pass of the network on state $s_t$ now outputs the value function $V(s_t)$ and the confidence the network has in its value function prediction, $c(s_t)$. Figure 1 shows the CARA3C network unrolled over time and it visually demonstrates how the confidence values are calculated using the network. Next, using eqs. (1) to (5) the weighted average of $n$-step returns is calculated and used as a target for improving $V(s_t)$. Algorithm 1, in Appendix $F$, presents the detailed pseudo-code for training a CARA3C agent. The policy improvement is carried out by following sample estimates of the loss function's gradient, given by: $\nabla_{\theta_a} \log \pi_{\theta_a}(a_t|s_t)\delta_t$, where $\delta_t$ is now defined in terms of the TD error term obtained by using *autodidactic returns* as the TD-target. Overall, the sample estimates for the gradient of the actor and the critic loss functions respectively are:

$$\nabla_{\theta_a} \log \pi_{\theta_a}(a_t|s_t)(G_t^{w(s_t)} - V(s_t))$$

$$\nabla_{\theta_c}(G_t^{w(s_t)} - V_{\theta_c}(s_t))^2$$

### 3.5 AVOIDING PITFALLS IN TD LEARNING OF CRITIC

The LSTM-A3C neural networks for representing the policy and the value function share all but the last output layer. In specific, the LSTM (Hochreiter & Schmidhuber, 1997) controller which aggregates the observations temporally is shared by the policy and the value networks. As stated in the previous sub-section, we extend the A3C network to predict the confidence values by creating

a new output layer which takes as input the LSTM output vector (LSTM outputs are the pre-final layer). Figure 1 contains a demonstration of how $w(s_1)$ is computed. Since all the three outputs (policy, value function, confidence on value function) share all but the last layer, $G_t^{w(s_t)}$ depends on the parameters of the network which are used for value function prediction. Hence, the autodidactic returns also influence the gradients of the LSTM controller parameters. However, it was observed that when the TD target, $G_t^w$, is allowed to move towards the value function prediction $V(s_t)$, it makes the learning unstable. This happens because the $\mathcal{L}_2$ loss between the TD-target and the value function prediction can now be minimized by moving the TD-target towards erroneous value function predictions $V(s_t)$ instead of the other way round. To avoid this instability we ensure that gradients do not flow back from the confidence values computation's last layer to the LSTM layer's outputs. In effect, the gradient of the critic loss with respect to the parameters utilized for the computation of the autodidactic return can no longer influence the gradients of the LSTM parameters (or any of the previous convolutional layers). To summarize, during back-propagation of gradients in the A3C network, the parameters *specific* to the computation of the autodidactic return do not contribute to the gradient which flows back into the LSTM layer. This ensures that the parameters of the confidence network are learned while treating the LSTM outputs as fixed feature vectors. This entire scheme of not allowing gradients to flow back from the confidence value computation to the LSTM outputs has been demonstrated in Figure 1. The forward arrows depict the parts of the network which are involved in forward propagation whereas the backward arrows depict the path taken by the back-propagation of gradients.

## 4 EXPERIMENTAL SETUP AND RESULTS

We performed general game-play experiments with CARA3C and LRA3C on 22 tasks in the Atari domain. All the networks were trained for 100 million time steps. The hyper-parameters for each of the methods were tuned on a subset of four tasks: Seaquest, Space Invaders, Gopher and Breakout. The same hyper-parameters were used for the rest of the tasks. The baseline scores were taken from Sharma et al. (2017). All our experiments were repeated thrice with different random seeds to ensure that our results were robust to random initialization. The same three random seeds were used across experiments and all results reported are the average of results obtained by using these three random seeds. Since the A3C scores were taken from Sharma et al. (2017), we followed the same training and testing regime as well. Appendix $A$ contains experimental details about the training and testing regimes. Appendix $G$ documents the procedure we used for picking important hyper-parameters for our methods.

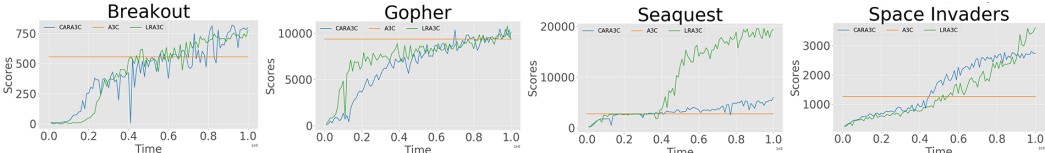

Figure 2: Training curves for raw scores obtained by CARA3C and LRA3C.

### 4.1 GENERAL GAMEPLAY PERFORMANCE

Table 1: Mean and median of A3C normalized scores across all games.

| Algorithm | Normalized Scores | |
| --- | --- | --- |
| | **Mean** | **Median** |
| A3C | 1.00 | 1.00 |
| LRA3C | 1.63 | 1.05 |
| CARA3C | **4.53** | **1.15** |

Evolution of the average performance of our methods with training progress has been shown in Figure 2. An expanded version of the graph for all the tasks can be found in Appendix $C$. Table 1

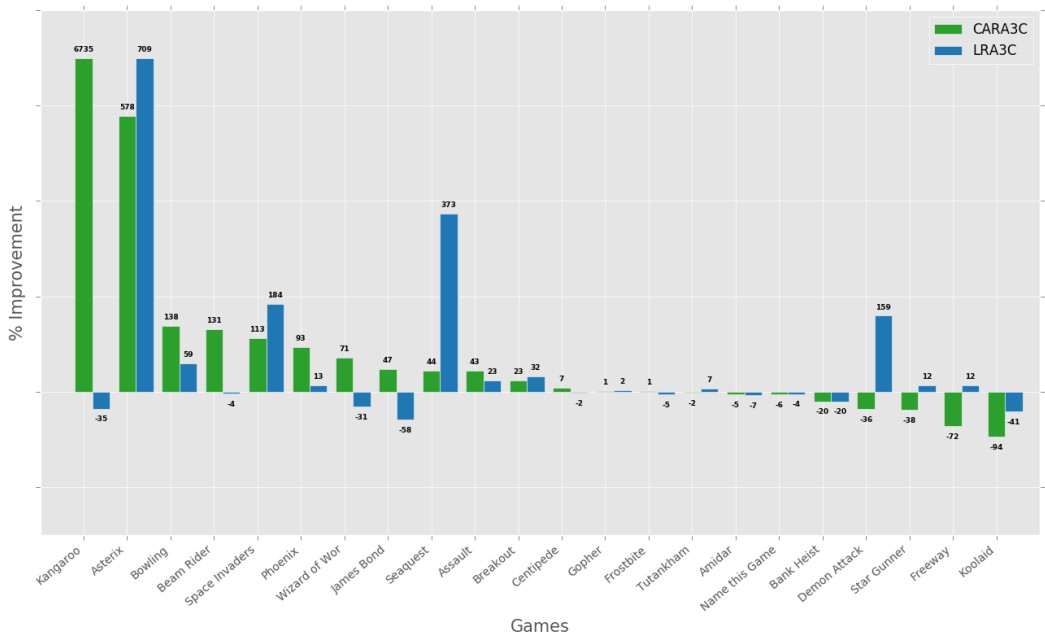

Figure 3: Percentage improvement achieved by CARA3C and LRA3C over A3C.

shows the mean and median of the A3C normalized scores of CARA3C and LRA3C. If the scores obtained by one of the methods and A3C in a task are $p$ and $q$ respectively, then the A3C normalized score is calculated as: $\left(\frac{p}{q}\right)$. As we can see, both CARA3C and LRA3C improve over A3C with CARA3C doing the best: on an average, it achieves over $4\times$ the scores obtained by A3C. The raw scores obtained by our methods against A3C baseline scores can be found in Table 2 (in Appendix $B$). Figure 3 shows the percentage improvement achieved by sophisticated mixture of $n$-step return methods (CARA3C and LRA3C) over A3C. If the scores obtained by one of the methods and A3C in a task are $p$ and $q$ respectively, then the percentage improvement is calculated as: $\left(\frac{p-q}{q} \times 100\right)$. As we can see, CARA3C achieves a staggering $67\times$ performance in the task Kangaroo.

## 4.2 ANALYSIS OF DIFFERENCE IN WEIGHTS ASSIGNED BY CARA3C AND LRA3C

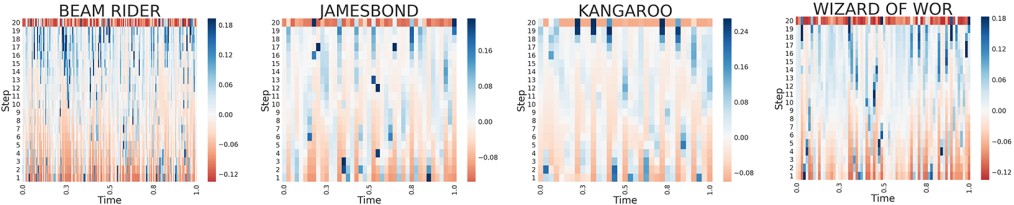

Figure 4: Difference in weights given by CARA3C and LRA3C to each of the n-step returns (where $n \leq 20$) during an episode.

Figure 4 demonstrates the difference in weights assigned by CARA3C and LRA3C (i.e $w^{(i)}_{CARA3C} - w^{(i)}_{LRA3C}$) to various $n$-step returns over the duration of an episode by fully trained DRL agents. The four games shown here are games where CARA3C achieves large improvements in performance over LRA3C. It can be seen that for all the four tasks, the difference in weights evolve in a dynamic fashion as the episode goes on. The motivation behind this analysis is to understand how different the weights used by CARA3C are as compared to LRA3C and as it can be seen, it's very different. The agent is clearly able to weighs it's returns in a way that is very different from LRA3C and this, in fact, seems to give it the edge over both LRA3C and A3C in many games. These results once again

reiterates the motivation for using dynamic Autodidactic Returns. An expanded version of the graphs for all the tasks can be found in Appendix $D$.

## 4.3 SIGNIFICANCE OF THE CONFIDENCE VALUES

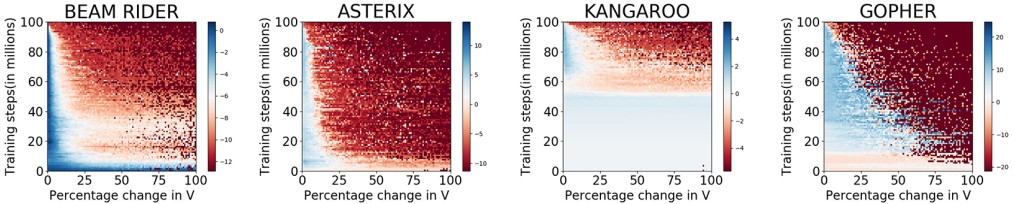

Figure 5: Relation between the confidence assigned to a state and the percent change in their value estimate. Percentage change in value estimates were obtained by calculating the value estimate of a state just before and after a batch gradient update step.

In Figure 5 each bin $(x, y)$ denotes the average confidence value assigned by the network to the states that were encountered during the training time between $y$ million and $y + 1$ million steps and whose value estimates were changed by a value between $x$ and $x + 1$ percent, where $0 \leq x < 100$ and $0 \leq y < 100$. In all the graphs we can see that during the initial stages of the training(lower rows of the graph), the confidence assigned to the states is approximately equal irrespective of the change is value estimate. As training progresses the network learns to assign relatively higher confidence values to the states whose value function changes by a small amount than the ones whose value function changes more. So, the confidence value can be interpreted as a value that quantifies the certainty the network has on the value estimate of that state. Thus, weighing the n-step returns based on the confidence values will enable the network to bootstrap the target value better. The confidence value depends on the certainty or the change in value estimate and it is not the other way round, i.e., having high confidence value doesn't make the value estimate of that state to change less. This is true because the confidence value can not influence the value estimate of a state as the gradients obtained from the confidence values are not back propagated beyond the dense layer of confidence output.

## 4.4 ANALYSIS OF EVOLUTION OF CONFIDENCE VALUES DURING AN EPISODE

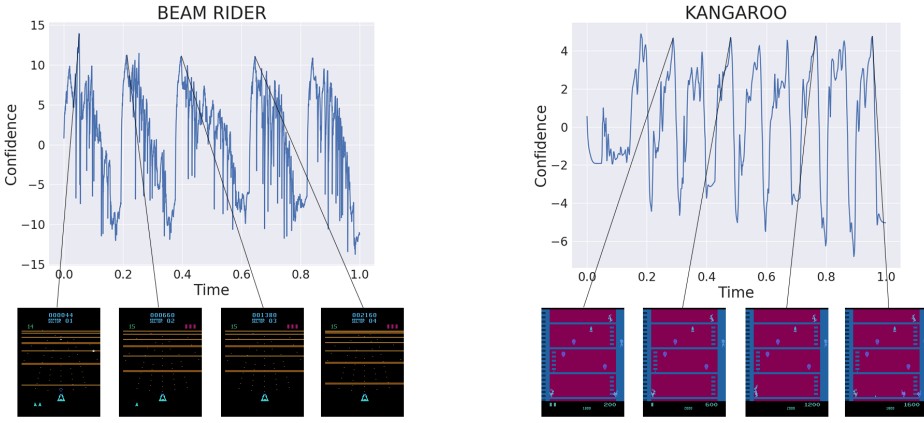

Figure 6: Evolution of confidence over an episode along with game frames for certain states with high confidence values.

Figure 6 shows the confidence values assigned to states over the duration of an episode by a fully trained CARA3C DRL agent for two games where CARA3C achieves large improvements over LRA3C and A3C: Kangaroo and Beam Rider.

For both the games it's clear that the confidence values change dynamically in response to

the states encountered. Here, it's important to note that the apparent periodicity observed in the graphs is not because of the nature of the confidences learnt but is instead due to the periodic nature of the games themselves. From the game frames shown one can observe that the frames with high confidence are highly recurring key states of the game. In case of Beam Rider, the high confidence states correspond to the initial frames of every wave wherein a new horde of enemies (often similar to the previous wave) come in after completion of the penultimate wave in the game. In the case of Kangaroo, apart from many other aspects to the game, there is piece of fruit which keeps falling down periodically along the left end of the screen (can be seen in the game's frames). Jumping up at the appropriate time and punching the fruit gives you 200 points. By observing game-play, we found that the policy learnt by the CARA3C agent identifies exactly this facet of the game to achieve such large improvements in the score. Once again, these series of states encompassing this transition of the fruit towards the bottom of the screen form a set of highly recurring states, Especially states where the piece of fruit is "jumping-distance" away the kangaroo and hence are closer to the reward are found to form the peaks. These observations reiterate the results obtained in Section 4.3 as the highly recurring nature of these states (during training over multiple episodes) would enable the network to estimate these value functions better. The better estimates of these value functions are then suitably used to obtain better estimates for other states by bootstrapping with greater attention to the high confidence states. We believe that this ability to derive from a few key states by means of an attention mechanism provided by the confidence values enables CARA3C to obtain better estimates of the value function. An expanded version of the graphs for all the tasks can be found in Appendix $E$.

## 4.5 ANALYSIS OF THE LEARNED VALUE FUNCTION

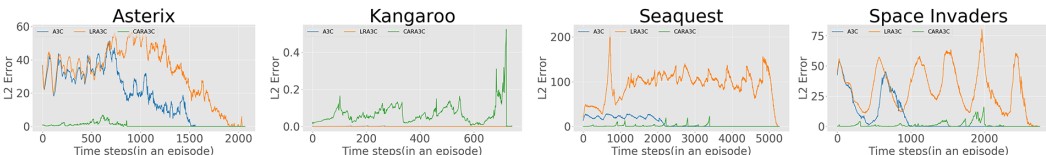

Figure 7: Comparison of value function estimates of CARA3C, LRA3C and A3C.

In this paper, we propose two methods for learning value functions in a more sophisticated manner than using $n$-step returns. Hence, it is important to analyze the value functions learned by our methods and understand whether our methods are indeed able to learn better value functions than baseline methods or not. For this sub-section we trained a few A3C agents to serve as baselines. To verify our claims about better learning of value functions, we conducted the following experiment. We took trained CARA3C, LRA3C and A3C agents and computed the $\mathcal{L}_2$ loss between the value function $V(s_t)$ predicted by a methods and the actual discounted sum of returns ($\sum_{k=0}^{T-t} \gamma^k r_{t+k}$). We averaged this quantity over 10 episodes and plotted it as a function of time steps within an episode. Figure 7 demonstrates that our novel method CARA3C learns a much better estimate of the Value function $V(s_t)$ than LRA3C and A3C. The only exception to this is the game of Kangaroo. The reason that A3C and LRA3C critics manage to estimate the value function well in Kangaroo is because the policy is no better than random and in fact their agents often score just around 0 (which is easy to estimate).

## 5 CONCLUSION AND FUTURE WORK

We propose a straightforward way to incorporate $\lambda$-returns into the A3C algorithm and carry out a large-scale benchmarking of the resulting algorithm LRA3C. We go on to propose a natural generalization of $\lambda$-returns called Confidence-based Autodidactic returns (CAR). In CAR, the agent learns to assign weights dynamically to the various $n$-step returns from which it can bootstrap. Our experiments demonstrate the efficacy of sophisticated mixture of multi-steps returns with at least one of CARA3C or LRA3C out-performing A3C in 18 out of 22 tasks. In 9 of the tasks CARA3C performs the best whereas in 9 of them LRA3C is the best. CAR gives the agent the freedom to learn and decide how much it wants to weigh each of its $n$-step returns.

The concept of Autodidactic Returns is about the generic idea of giving the DRL agent the ability to model confidence in its own predictions. We demonstrate that this can lead to better

TD-targets, in turn leading to improved performances. We have proposed only one way of modeling the autodidactic weights wherein we use the confidence values that are predicted alongside the value function estimates. There are multiple other ways in which these $n$-step return weights can be modeled. We believe these ways of modeling weighted returns can lead to even better generalization in terms how the agent perceives it's TD-target. Modeling and bootstrapping off TD-targets is fundamental to RL. We believe that our proposed idea of CAR can be combined with any DRL algorithm (Mnih et al., 2015; Jaderberg et al., 2017; Sharma et al., 2017) wherein the TD-target is modeled in terms of $n$-step returns.

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

## APPENDIX A: EXPERIMENTAL DETAILS

Since the baseline scores used in this work are from Sharma et al. (2017), we use the same training and evaluation regime as well.

### ON HYPER-PARAMETERS

We used the LSTM-variant of A3C [Mnih et al. (2016)] algorithm for the CARA3C and LRA3C experiments. The async-rmsprop algorithm [Mnih et al. (2016)] was used for updating parameters with the same hyper-parameters as in Mnih et al. (2016). The initial learning rate used was $10^{-3}$ and it was linearly annealed to 0 over 100 million time steps, which was the length of the training period. The $n$ used in $n$-step returns was 20. Entropy regularization was used to encourage exploration, similar to Mnih et al. (2016). The $\beta$ for entropy regularization was found to be 0.01 after hyper-parameter tuning, both for CARA3C and LRA3C, separately. The $\beta$ was tuned in the set $\{0.01, 0.02\}$. The optimal initial learning rate was found to be $10^{-3}$ for both CARA3C and LRA3C separately. The learning rate was tuned over the set $\{7 \times 10^{-4}, 10^{-3}, 3 \times 10^{-3}\}$. The discounting factor for rewards was retained at 0.99 since it seems to work well for a large number of methods (Mnih et al., 2016; Sharma et al., 2017; Jaderberg et al., 2017). The most important hyper-parameter in the LRA3C method is the $\lambda$ for the $\lambda$-returns. This was tuned extensively over the set $\{0.05, 0.15, 0.5, 0.85, 0.9, 0.95, 0.99\}$. The best four performing models have been reported in Figure 11b. The best performing models had $\lambda = 0.9$.

All the models were trained for 100 million time steps. This is in keeping with the training regime in Sharma et al. (2017) to ensure fair comparisons to the baseline scores. Evaluation was done after every 1 million steps of training and followed the strategy described in Sharma et al. (2017) to ensure fair comparison with the baseline scores. This evaluation was done after each 1 million time steps of training for 100 episodes , with each episode's length capped at 20000 steps, to arrive at an average score. The evolution of this average game-play performance with training progress has been demonstrated for a few tasks in Figure 2. An expanded version of the figure for all the tasks can be found in Appendix $C$.

Table 2 in Appendix $B$ contains the raw scores obtained by CARA3C, LRA3C and A3C agents on 22 Atari 2600 tasks. The evaluation was done using the *latest* agent obtained after training for 100 million steps, to be consistent with the evaluation regime presented in Sharma et al. (2017) and Mnih et al. (2016).

### ARCHITECTURE DETAILS

We used a low level architecture similar to Mnih et al. (2016); Sharma et al. (2017) which in turn uses the same low level architecture as Mnih et al. (2015). Figure 1 contains a visual depiction of the network used for CARA3C. The common parts of the CARA3C and LRA3C networks are described below.

The first three layers of both the methods are convolutional layers with same filter sizes, strides, padding and number of filters as Mnih et al. (2015; 2016); Sharma et al. (2017). These convolutional layers are followed by two fully connected (FC) layers and an LSTM layer. A policy and a value function are derived from the LSTM outputs using two different output heads. The number of neurons in each of the FC layers and the LSTM layers is 256. These design choices have been taken from Sharma et al. (2017) to ensure fair comparisons to the baseline A3C model and apply to both the CARA3C and LRA3C methods.

Similar to Mnih et al. (2016) the Actor and Critic share all but the final layer. In the case of CARA3C, Each of the three functions: policy, value function and the confidence value are realized with a different final output layer, with the confidence and value function outputs having no non-linearity and one output-neuron and with the policy and having a softmax-non linearity of size equal to size of the action space of the task. This non-linearity is used to model the multinomial distribution.

APPENDIX B: TABLE OF RAW SCORES

Table 2: Game Playing Experiments on Atari 2600

| Name | CARA3C | LRA3C | A3C |
|------|--------|-------|-----|
| Amidar | 973.73 | 954.79 | **1028.34** |
| Assault | **2670.74** | 2293.74 | 1857.61 |
| Asterix | 16048.00 | **19139.00** | 2364.00 |
| Bank Heist | 1378.60 | 1382.97 | **1731.40** |
| Beam Rider | **5076.10** | 2099.71 | 2189.96 |
| Bowling | **40.94** | 26.90 | 16.88 |
| Breakout | 678.06 | **733.87** | 555.05 |
| Centipede | **3553.82** | 3226.94 | 3293.33 |
| Demon Attack | 17065.22 | **69373.42** | 26742.75 |
| Freeway | 4.9 | **19.97** | 17.68 |
| Frostbite | **310.00** | 289.27 | 306.8 |
| Gopher | 9468.07 | **9626.53** | 9360.60 |
| James Bond | **421.50** | 118.33 | 285.5 |
| Kangaroo | **1777.33** | 16.67 | 26.00 |
| Koolaid | 66.67 | 663.67 | **1136** |
| Name this game | 11354.37 | 11515.33 | **12100.80** |
| Phoenix | **10407.83** | 6084.73 | 5384.1 |
| Sea quest | 4056.4 | **13257.73** | 2799.60 |
| Space Invaders | 2703.83 | **3609.65** | 1268.75 |
| Star Gunner | 24550.33 | **44942.33** | 39835.00 |
| Tutankhamun | 245.86 | **271.46** | 252.82 |
| Wizard of Wor | **5529.00** | 2205.67 | 3230.00 |

All the evaluations were done using the agent obtained after training for 100 million steps, to be consistent with the evaluation paradigm presented in Sharma et al. (2017) and Mnih et al. (2016). Both CARA3C and LRA3C scores are obtained by averaging across 3 random seeds. The scores for A3C column were taken from Table 4 of Sharma et al. (2017).

APPENDIX C: TRAINING GRAPHS

The evaluation strategy described in Appendix $A$ was executed to generate training curves for all the 22 Atari tasks. This appendix contains all those training curves. These curves demonstrate how the performance of the CARA3C and LRA3C agents evolves with time.

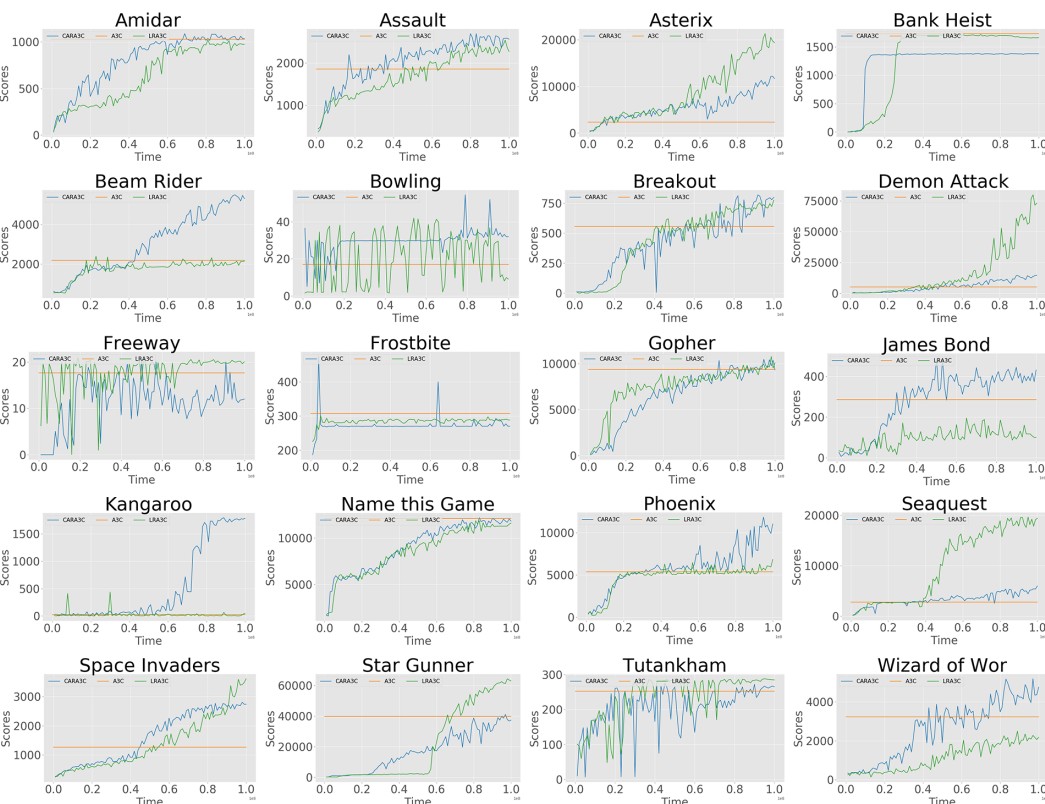

Figure 8: Training curves for CARA3C and LRA3C

APPENDIX D: DIFFERENCE IN WEIGHTS ASSIGNED BY CARA3C AND LRA3C

This appendix presents the expanded version of the results shown in Section 4.2. These plots show the vast differences in weights assigned by CARA3C and LRA3C to various $n$-step returns over the duration of a single episode.

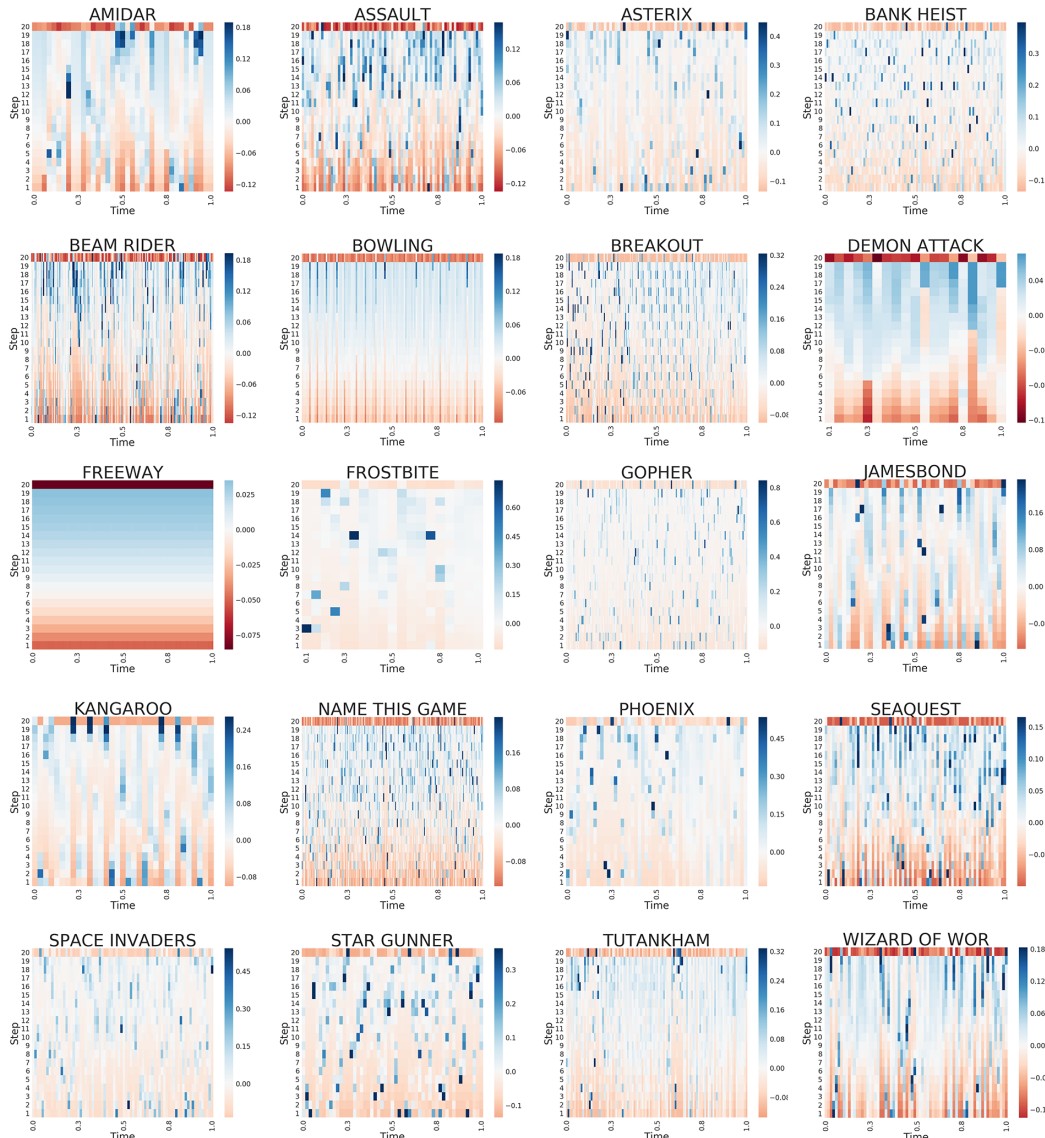

Figure 9: Difference in weights given by CARA3C and LRA3C to each of the n-step returns (where $n \leq 20$) during an episode.

## APPENDIX E: EVOLUTION OF CONFIDENCE VALUES DURING AN EPISODE

This appendix presents the expanded version of the results shown in Section 4.4. The aim is to show the how the confidence values dynamically change over the duration of an episode and show the presence of explicit peaks and troughs in many games. Here, it's important to note that the apparent periodicity observed in some graphs is not because of the nature of the confidences learnt but is instead due to the periodic nature of the games themselves.

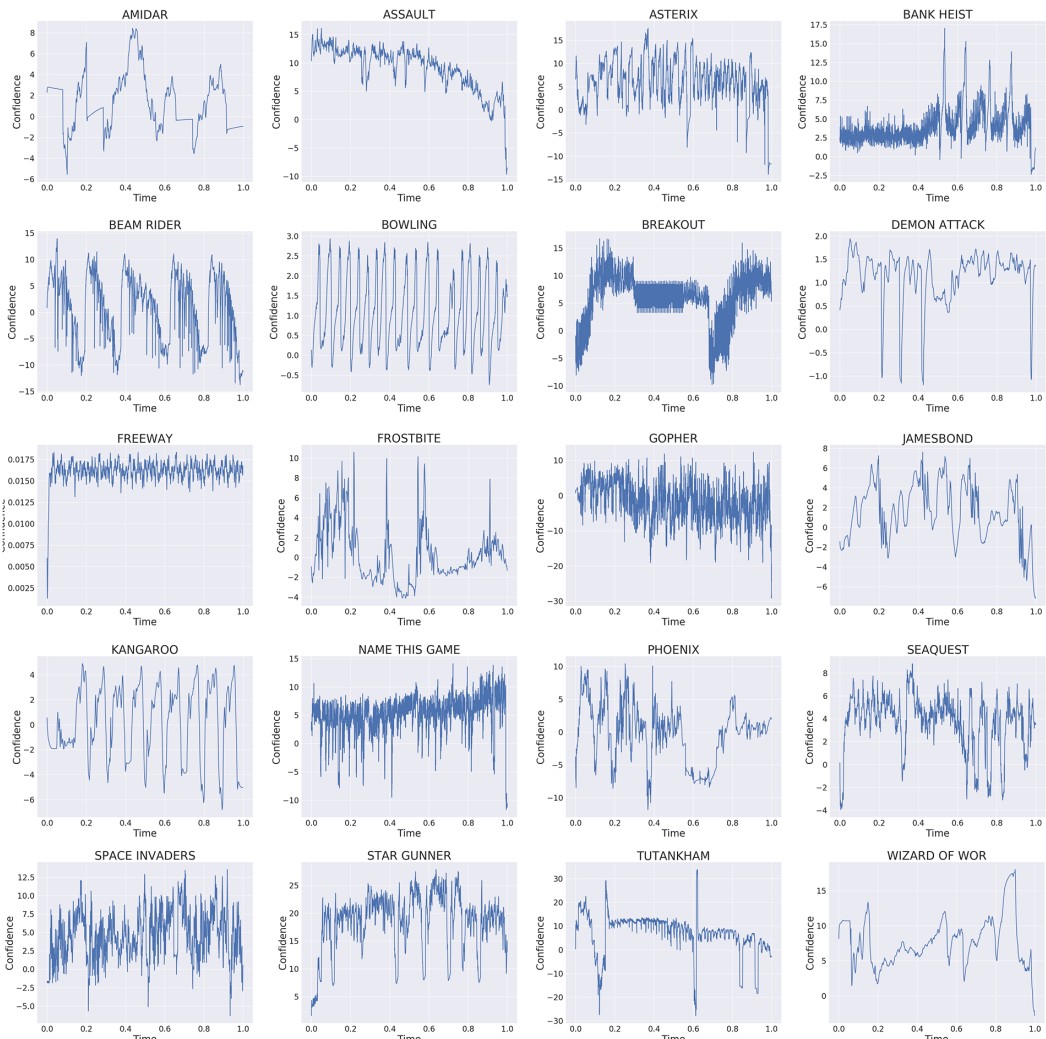

Figure 10: Evolution of confidence values over an episode.

## APPENDIX F: ALGORITHM FOR TRAINING CARA3C

The algorithm corresponding to CARA3C, our main contribution, has been presented in Algorithm 1. The get_weights_matrix function used by this algorithm in line 30 is given by Algorithm 2. A similar algorithm can be constructed for LRA3C easily.

---

**Algorithm 1** CARA3C

---

1: // Assume global shared parameter vectors $\theta$.
2: // Assume global step counter (shared) $T = 0$
3:
4: $K \leftarrow$ Maximum value of $n$ in $n$-step returns
5: $T_{\max} \leftarrow$ Total number of training steps for CARA3C
6: $\pi \leftarrow$ Policy of the agent
7: Initialize local thread's step counter $t \leftarrow 1$
8: Let $\theta'$ be the local thread's parameters
9:
10: **repeat**
11:     $t_{init} = t$
12:     $d\theta \leftarrow 0$
13:     $dw \leftarrow 0$
14:     Synchronize local thread parameters $\theta' = \theta$
15:     confidences $\leftarrow$[ ]
16:     Obtain state $s_t$
17:     states $\leftarrow$ [ ]
18:     **repeat**
19:         states.append($s_t$)
20:         Sample $a_t \sim \pi(a_t|s_t; \theta')$
21:         Execute action $a_t$ to obtain reward $r_t$ and obtain next state $s_{t+1}$
22:         confidences.append($c(s_{t+1}; \theta')$)
23:         $t \leftarrow t + 1$
24:         $T \leftarrow T + 1$
25:     **until** $s_t$ is terminal or $t == t_{init} + K$
26:     **if** $s_t$ is terminal **then**
27:         $R \leftarrow 0$
28:     **else**
29:         $R \leftarrow V(s_t; \theta')$
30:     $C \leftarrow$ get_weights_matrix(confidences)
31:     $G_{ij} \leftarrow$ the $i$-step return used for bootstraping estimate for states[$j$]     // $K \times K$ matrix
32:     **for** $i \in \{t - 1, \cdots, t_{init}\}$ **do**
33:         $j \leftarrow i - t_{init}$
34:         $R' \leftarrow R$
35:         **for** $k \in \{K - 1, \cdots, j\}$ **do**
36:             $R' \leftarrow r_i + \gamma R'$
37:             $G_{kj} \leftarrow C_{kj}.R'$   // Assuming 0-based indexing
38:         $R \leftarrow V(s_i; \theta')$
39:
40:     **for** $i \in \{t - 1, \ldots t_{init}\}$ **do**
41:         $j \leftarrow i - t_{init}$
42:         TD-target $\leftarrow \sum_{j=0}^{K-1} G_{ij}$
43:         Gradients for $\theta$ based on $\pi$: $d\theta' \leftarrow d\theta' + \nabla_\theta \Big( log(\pi(a_i|s_i; \theta')) \Big) \Big( \text{TD-target} - V(s_i) \Big)$
44:         Gradients for $\theta$ based on $V$: $d\theta' \leftarrow d\theta' + \nabla_\theta \Big( \text{TD-target} - V(s_i; \theta') \Big)^2$
45:     Perform asynchronous update of $\theta$ using $d\theta'$
46: **until** $T > T_{max}$

---

---

**Algorithm 2** Creates a 2D weights matrix with the confidence numbers

---

    **function** GET_WEIGHTS_MATRIX($C$)
2:      $w \leftarrow \text{softmax}(C)$
      $W \leftarrow \mathbf{1} \otimes w^T$    // $W$ is a $K \times K$ weight matrix with $W_i = w^T$. $\otimes$ is outer product.
4:      **for** $w_{i,j} \in W$ **do**
          **if** $i > j$ **then**
6:            $w_{i,j} \leftarrow 0$
      **for** $w_i \in W$ **do**
8:         $w_i = \frac{w_i}{\text{sum}(w_i)}$
      **return** $W$

---

## APPENDIX G: CHOICE OF IMPORTANT HYPER-PARAMETERS FOR OUR METHODS

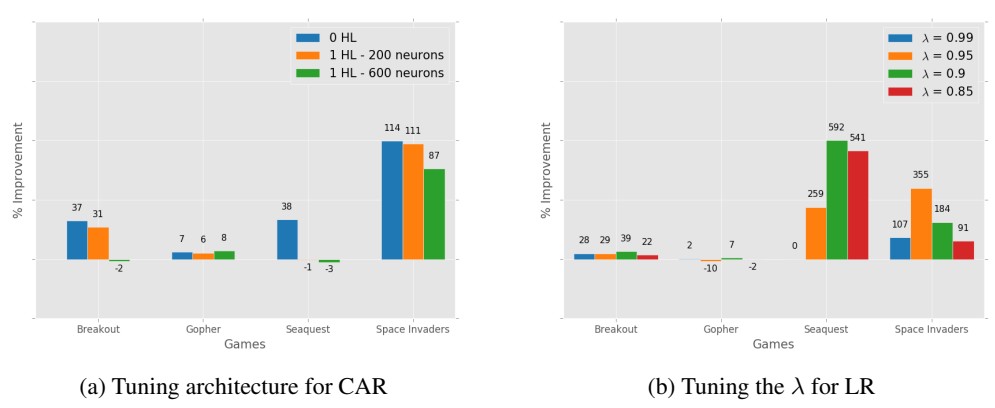

(a) Tuning architecture for CAR        (b) Tuning the $\lambda$ for LR

Figure 11: Tuning the important hyper-parameters for our methods

Perhaps the most important hyper-parameter in CARA3C is the network which calculates the confidence values. Since gradients do not flow back from confidence computation to the LSTM controller, this becomes an important design choice. We experimented extensively with different types of confidence computation networks including shallow ones, deep ones, wide ones and narrow ones. We found a "zero hidden layer" network on top of the LSTM controller (much like one which computes the value function) works the best (Figure 11a). Similarly, the most important hyper-parameter in $\lambda$-returns is the $\lambda$ from eq. (3). While we experimented with a large number and range of values for $\lambda$ the best performing ones have been reported in Figure 11b.

