# OpenReview forum: "Learning to Mix n-Step Returns: Generalizing Lambda-Returns for Deep Reinforcement Learning"
_ICLR.cc/2018/Conference — Reject_

### Official Review · AnonReviewer3 · 2017-11-22
**The paper extends the A3C algorithm with lambda returns.  In addition it proposes  an approach for learning the weights of the returns.**

**Rating:** 5
**Confidence:** 3

**Review:**

SUMMARY
The major contribution of the paper is a generalization of lambda-returns called Confidence-based Autodidactic Returns (CAR), wherein the RL agent learns the weighting of the n-step returns in an end-to-end manner.   These CARs are used in the A3C algorithm.  The weights are based on the confidence of the value function of the n-step return.  Even though this idea in not new the authors propose a simple and robust approach for doing it by using the value function estimation network of A3C.

During experiments, the autodidactic returns perform better only half of the time as compared to lambda returns.


COMMENTS
The j-step returns TD error is not written correctly

In Figure 1 it is not obvious how the confidence of the values is estimated.
Figure 1 is unreadable.


A lighter version of Algorithm 1 in Appendix F should be moved in the text, since this is the novelty of the paper.

---

> ### Author Response · Authors · 2018-01-05
> **Regarding points raised by Reviewer 3**
>
> We thank the reviewer for the valuable feedback.
>
> 1) LRA3C is the best algorithm in 9 out of 22 games and CARA3C is the best in 9 other games. In total, sophisticated mixtures of multi-step return methods perform better in 18 out of 22 games. This supports one of our claims that “sophisticated mixtures of multi-step return methods like lambda-returns and Confidence-based Autodidactic Returns leads to considerable improvement in the performance of a DRL agent”. We agree that CARA3C does not always outperform LRA3C. A binary comparison does seem to put both methods in an equal footing. But Table 1, shows that the average and median improvement in scores is clearly better for CARA3C compared to LRA3C. We believe that our work presents a new direction for research in value function learning to proceed. Due to limitation of computational  resources we have access to and time, we were able to evaluate CARA3C only in the Atari domain. But considering that estimation of value function is fundamental to RL, our work is widely applicable. Our analysis of the confidence values show that the learned confidence values are indeed non-trivial and enables the agent to dynamically weigh the n-step returns (Section 4.2). Section 4.4 even shows certain instances where this has clearly enables to agent to achieve better game play.  Our work proposes a simple and robust method for generalising the idea behind weighted returns. With all these in mind, we feel evaluation of our work based on the underlying theory, concept and it’s fundamentality is more reasonable at this stage rather than just based on it’s single-scenario performance in a limited domain such as Atari 2600 games.
>
> 2) This is a typo. We have fixed it in the revised version.
>
> 3) For predicting the confidence values, a distinct neural network is created which shares all but the last layer with the value function estimation network. So, every forward pass of the network on state s_t now outputs the value function V(s_t) and the confidence c(s_t). This is indicated in Figure 1. Figure 1 shows the CARA3C network unrolled over time and it visually demonstrates how the confidence values are calculated using the network for s_1. This figure demonstrates for m=20 and thus unrolls the network till s_21. So now we have V(s_2), V(s_3).......till V(s_21) and C(s_2), C(s_3).......till C(s_21). Using eqn(5), we now compute w(s_1)(i) for i=1 to i=20 and using eqn(2), we compute G_1(i) for 1=1 to i=20. Finally using eqn(1), we compute G_1(w) and this serves as an estimate for V(s_1). This completes the computation of the confidence values and how they are used. Now for learning the parameters we use back-propagation. However, during back-propagation of gradients in the A3C network, the parameters specific to the computation of the autodidactic return do not contribute to the gradient which flows back into the LSTM layer. This ensures that the parameters of the confidence network are learned while treating the LSTM outputs as fixed feature vectors. This entire scheme of not allowing gradients to flow back from the confidence value computation to the LSTM outputs has been demonstrated in Figure 1. The forward arrows depict the parts of the network which are involved in forward propagation whereas the backward arrows depict the path taken by the back-propagation of gradients.  The complete reasoning behind this can be found in Section 3.5.
>
> 4) For the legibility of the image, we have ensured that the image has a high resolution and we feel that the elements are quite clear when one zooms in. However, we will add a larger image in the camera-ready version.
>
> 5) Thanks for raising this point. We will add a concise version of the algorithm in the main paper for our camera-ready version.

---

### Official Review · AnonReviewer2 · 2017-11-27
**An interesting adaptive weight method for using eligibility vectors with deep RL. I am a little concerned that a key part of the method is imperfectly described.**

**Rating:** 6
**Confidence:** 4

**Review:**

The authors present confidence-based autodidactic returns, a Deep learning RL method to adjust the weights of an eligibility vector in TD(lambda)-like value estimation to favour more stable estimates of the state. The key to being able to learn these confidence values is to not allow the error of the confidence estimates propagate back though the deep learning architecture.

However, the method by which these confidence estimates are refined could be better described. The authors describe these confidences variously as: "some notion of confidence that the agent has in the value function estimate" and "weighing the returns based on a notion of confidence has been explored earlier (White & White, 2016; Thomas et al., 2015)". But the exact method is difficult to piece together from what is written. I believe that the confidence estimates are considered to be part of the critic and the w vector to be part of the theta_c parameters. This would then be captured by the critic gradient for the CAR method that appears towards the end of page 5. If so, this should be stated explicitly.

There is another theoretical point that could be clearer. The variation in an autodidactic update of a value function (Equation (4)) depends on a few things, the in variation future value function estimates themselves being just one factor. Another two sources of variation are: the uncertainty over how likely each path is to be taken, and the uncertainty in immediate rewards accumulated as part of some n-step return. In my opinion, the quality of the paper would be much improved by a brief discussion of this, and some reflection on what aspects of these variation contribute to the confidence vectors and what isn't captured.

Nonetheless, I believe that the paper represents an interesting and worthy submission to the conference. I would strongly urge the authors to improve the method description in the camera read version though. A few additional comments are as follows:

  • The plot in Figure 3 is the leading collection of results to demonstrate the dominance of the authors' adaptive weight approach (CAR) over the A3C (TD(0) estimates) and LRA3C (truncated TD(lambda) estimates) approaches. However, the way the results are presented/plotted, namely the linear plot of the (shifted) relative performance of CAR (and LRA3C) versus A3C, visually inflates the importance of tasks on which CAR (and LRA3C) perform better than A3C, and diminishes the importance of those tasks on which A3C performs better. It would be better kept as a relative value and plotted on a log-scale so that positive and negative improvements can be viewed on an equal setting.
  • On page 3, when Gt is first mentioned, Gt should really be described first, before the reader is told what it is often replaced with.
  • On page 3, where delta_t is defined (the j step return TD error, I think the middle term should be $gamma^j V(S_{t+j})$
  • On page 4 and 5, when describing the gradient for the actor and critic, it would be better if these were given their own terminology, but if not, then use of the word respectively in each case would help.

---

> ### Author Response · Authors · 2018-01-05
> **Regarding points raised by Reviewer 2**
>
> We thank the reviewer for the generally positive review and valuable feedback.
>
> 1) This is correct. To be clear, as shown in Figure 1, the network outputs the value function, confidence and policy. The additional output of confidence is our primary modification to the original A3C network. Specifically, the LSTM controller which aggregates the observations temporally is shared by the policy and the value networks. We extend the A3C network to predict the confidence values by creating a new output layer which takes as input the LSTM output vector (LSTM outputs are the pre-final layer).  The confidence estimate are used to compute G_t^{w}(s_t) and the LSTM-to-Confidence parameters (confidence network)  are updated using the gradient from the critic. But as described in Section 3.5, we do not allow these gradients to flow back from the confidence values computation last layer to the LSTM layer’s outputs. The parameters of the confidence network are learned while treating the LSTM outputs as fixed feature vectors. It’s however important to keep in mind that since all the three outputs (policy, value function, confidence on value function) share all but the last layer,  G_t^{w}(s_t) depends on the parameters of the network which are used for value function prediction. But the gradients arising from the confidence estimates are used to update only newly added confidence network parameters and not the shared parameters.
>
> All these points have already been mentioned in the paper (spread across sections 3.2, 3.4 and 3.5). However, we hope this summarization picks out the relevant points and provides some clarity.
>
> 2) The variation in the reward ​is an "irreducible" component, and as such would not play a role in the variation of the confidence scores. The variation in the path taken is also irreducible in some sense, but can lead to larger variation in the estimates. But even this would average out in the long run. What would happen is that states that have highly uncertain trajectories after that would always have lower confidence scores. There is an irreducible level of uncertainty in the environment and we believe in the long run this will divert the agent into regions of the state where the irreducible variance or "risk" is lower. We agree that these points however require further analysis. Our current attempts at understanding the confidence values are however presented in Sections 4.2, 4.3 and 4.4 and we believe that it definitely provides some preliminary insights.
>
> 3) Figure: https://imgur.com/a/ZjHE1
> Our goal behind the plot was to use A3C as a baseline and show the improvements achieved by the CARA3C and LRA3C. With that in mind, showing improvement % seemed to be more appropriate. But please refer to the figure above where we plotted the relative value in log-scale. Though this version of the plot does also highlight games where A3C performs better, our claim that CARA3C and LRA3C does significantly better on an overall is still clearly visible.
>
> 4) We have introduced G_t first in Section 2.1 before using it in Page 3.
>
> 5) This is a typo. We have fixed it in the revised  version.
>
> 6) We didn’t realise this would lead to a confusion. We have fixed it in the revised version.

---

### Official Review · AnonReviewer1 · 2017-11-27
**Alternative definitions for lambda returns**

**Rating:** 5
**Confidence:** 4

**Review:**

This paper revisits the idea of exponentially weighted lambda-returns at the heart of TD algorithms. The basic idea is that instead of geometrically weighting the n-step returns we should instead weight them according to the agent's own estimate of its confidence in it's learned value function. The paper empirically evaluates this idea on Atari games with deep non-linear state representations, compared to state-of-the-art baselines.

This paper is below the threshold because there are issues with the : 1) motivation, 2) the technical details, and (3) the empirical results.

The paper begins by stating that the exponential weighting of lambda returns is ad-hoc and unjustified. I would say the idea is well justified in several ways. First the lambda return definition lends itself to online approximations that achieve a fully incremental online form with linear computation and nearly as good performance of the off-line version. Second, decades of empirical results illustrating good performance of TD compared with MC methods. And an extensive literature of theoretical results. The paper claims that the exponential has been noted to be ad-hoc, please provide a reference for this.

There have been several works that have noted that lambda can and perhaps should be changed as a function of state (Sutton and Barto, White and White [1], TD-Gammon). In fact, such works even not that lambda should be related to confidence. The paper should work harder to motivate why adapting lambda as a function of state---which has been studied---is not sufficient.

I don't completely understand the objective. Returns with higher confidence should be weighted higher, according to the confidence estimate around the value function estimate as a function of state? With longer returns, n>>1, the role of the value function in the target is down-weighted by gamma^n---meaning its accuracy is of little relevance to the target. How does your formalism take this into account? The basic idea of the lambda return assumes TD targets are better than MC targets due to variance, which place more weight on shorter returns.

I addition I don't understand how learning confidence of the value function has a realizable target. We do not get supervised targets of the confidence of our value estimates. What is your network updating toward?

The work of Konidaris et al [1] is a more appropriate reference for this work (rather than the Thomas reference provided). Your paper does not very clearly different itself from Konidaris's work here. Please expand on this.

The experiments have some issues. One issue is that basic baselines could more clearly illustrate what is going on. There are two such baselines: random fixed weightings of the n-step returns, and persisting with the usual weighting but changing lambda on each time step (either randomly or according to some decay schedule). The first baseline is a sanity check to ensure that you are not observing some random effect. The second checks to see if your alternative weighting is simply approximating the benefits of changing lambda with time or state.

I would say the current results indicate the conventional approach to TD is working well if not better than the new one. Looking at fig 3, its clear the kangaroo is skewing the results, and that overall the new method is performing worse. This is further conflated by fig7 which attempts to illustrate the quality of the learned value functions. In Kangaroo, the domain where your method does best, the l2 error is worse. On the other hand in sea quest and space invaders, where your method does worse, the l2 error is better. These results seem conflicting, or at least raise more questions than they answer.

[1] A Greedy Approach to Adapting the Trace Parameter for Temporal Difference Learning . Adam White and Martha White. Autonomous Agents and Multi-agent Systems (AAMAS), 2016
[2] G. D. Konidaris, S. Niekum, and P. S. Thomas. TDγ: Re-evaluating complex backups in temporal difference learning. In Advances in Neural Information Processing Systems 24, pages 2402–2410. 2011.

---

> ### Author Response · Authors · 2018-01-05
> **Regarding points raised by Reviewer 1**
>
> We thank the reviewer for the valuable feedback.
>
> 1) Can the reviewer please point us to justifications in favour of exponential weighting of lambda-returns? We will try to find such a reference. However, this seems more like an opinion than a fact and hence we are also ok with removing that second statement.
>
> 2) Adapting lambda as a function of state is not sufficient for a simple reason: if the best next state (say s_{t+k}) for estimating the returns for current state (s_t) is not the immediately next one (s_{t+1}), no value of lambda can capture this.  Thus we wanted to move past this dependence on lambda and give the agent the complete freedom learn the importance of all i-step returns for n >= i >=1 for every state s, on it’s own. This is how the idea of autodidactic returns and eventually confidence-based autodidactic returns was formulated.
>
> 3) Comment on feasibility of n>>1:
> For n-step returns, it has been empirically observed [cite A3C] that it is in fact for an intermediate value of n (like n = 5) that best results are obtained. This is because of the bias-variance trade-off in estimating the returns with larger values of n favouring lower bias but higher variance.
>
> Our formalism:
> Our formalism allows the agent to learn the importance of all i-step returns for n >= i >=1 for every state s. This is regardless of how large n is. It is true that when n is large the down-weighted value function’s importance in the target is low, but this is reflected only indirectly in our formalism. Which is to say, if such returns (i-step returns for i >>1) are not beneficial to the learning problem, the agent will automatically learn small weights corresponding to them. If it doesn’t, this perhaps means that the variance in the empirical estimation of the return is low.
>
> 4) The network is updating the confidence values using the same objective function which is used for training the value function. We request the reviewer to go through second equation in section 3.4 and equation (5). Our answer for Reviewer 2’s first point may also provide more clarity.
>
> 5) We thank the reviewer for introducing us to  this work that we  were unfamiliar with. We however feel that the issues addressed in that paper are quite different from that of our work and we are not exactly sure where the reviewer wants us to differentiate. Can the reviewer please elaborate?
>
> 6) We believe the analysis presented in Sections 4.2, 4.3 and 4.4 sufficiently demonstrate the utility of our approach.  While the additional experiments would certainly help, we feel that our current results already show that CARA3C does learn something non-trivial and does not explicitly raise the necessity for such sanity checks.
>
> 7) Reviewer 3 seems to have a similar concern. We believe that our answer to his first point will address this as well.
>
> 8) The value function that we are using here is that corresponding to the policies learned by the agents. This in no way talks about the quality of the policy. We are only talking about the approximation of the value function of the policy learned by the agent. Since in Kangaroo the other agents learn uniformly zero value policies, it is easier to approximate and hence they do a better job. When the policies are non-trivial, then CARA3C does a much better job of approximating the value function.

---

### Decision · Program_Chairs · 2018-01-29
**ICLR 2018 Conference Acceptance Decision**

**Decision:**

Reject

**Comment:**

This is an interesting paper, but was quite difficult to follow. As they stand, the empirical results are not altogether convincing nor warrant acceptance.